

# Characterization of biofilm production in different strains of *Acinetobacter baumannii* and the effects of chemical compounds on biofilm formation

Ming-Feng Lin[1], Yun-You Lin[2] and Chung-Yu Lan[2,3]

[1] Department of Medicine, National Taiwan University Hospital Chu-Tung Branch, Hsinchu County, Taiwan
[2] Institute of Molecular and Cellular Biology, National Tsing Hua University, Hsinchu, Taiwan
[3] Department of Life Science, National Tsing Hua University, Hsinchu, Taiwan

Corresponding author
Ming-Feng Lin,
allenlmf2007@gmail.com

## ABSTRACT

*Acinetobacter baumannii*, an important emerging pathogen of nosocomial infections, is known for its ability to form biofilms. Biofilm formation increases the survival rate of *A. baumannii* on dry surfaces and may contribute to its persistence in the hospital environment, which increases the probability of nosocomial infections and outbreaks. This study was undertaken to characterize the biofilm production of different strains of *A. baumannii* and the effects of chemical compounds, especially antibiotics, on biofilm formation. In this study, no statistically significant relationship was observed between the ability to form a biofilm and the antimicrobial susceptibility of the *A. baumannii* clinical isolates. Biofilm formation caused by *A. baumannii* ATCC 17978 after gene knockout of two-component regulatory system gene *baeR*, efflux pump genes *emrA/emrB* and outer membrane coding gene *ompA* revealed that all mutant strains had less biofilm formation than the wild-type strain, which was further supported by the images from scanning electron microscopy and confocal laser scanning microscopy. The addition of amikacin, colistin, LL-37 or tannic acid decreased the biofilm formation ability of *A. baumannii*. In contrast, the addition of lower subinhibitory concentration tigecycline increased the biofilm formation ability of *A. baumannii*. Minimum biofilm eradication concentrations of amikacin, imipenem, colistin, and tigecycline were increased obviously for both wild type and multidrug resistant clinical strain *A. baumannii* VGH2. In conclusion, the biofilm formation ability of *A. baumannii* varied in different strains, involved many genes and could be influenced by many chemical compounds.

# INTRODUCTION

*Acinetobacter baumannii*, as an important emerging pathogen of nosocomial infection, is known for its ability to form biofilms (*Longo, Vuotto & Donelli, 2014*). Biofilm formation increases the survival rate of *A. baumannii* on dry surfaces and may contribute to its

persistence in the hospital environment, increasing the probability of causing nosocomial infections and outbreaks (*Espinal, Marti & Vila, 2012*).

The mechanisms to explain the increased drug resistance of bacteria related with biofilms are various and at least included delayed penetration of the antimicrobial agents into the biofilm and reduced growth rate of the microorganisms within the biofilm (*Donlan, 2000*). A positive correlation between biofilm formation and antimicrobial resistance in *A. baumannii* has been confirmed (*Badave & Kulkarni, 2015*), although one study suggested an inverse relationship between biofilm production and meropenem resistance in nosocomial *A. baumannii* isolates (*Perez, 2015*). The ability to form a biofilm may affect antibiotic susceptibility and clinical failure, even when the dose administered is in the susceptible range (*Kim et al., 2015*).

Modulation of biofilm formation is as diverse as the surface on which the bacteria reside and the cellular components involved in the programmed multi-step process (*Gaddy & Actis, 2009*). The regulation of biofilm formation is influenced by sensing bacterial cell density, the presence of different nutrients and the concentration of cations, which was through the two-component regulatory system BfmRS (*Luo et al., 2015a*). This transcriptional regulatory system activates the expression of the CsuA/BABCDE usher-chaperone assembly system responsible for the production of pili, which are needed for cell attachment and biofilm formation on polystyrene surfaces (*Tomaras et al., 2008*). Both biofilm-associated proteins and OmpA protein play a role in biofilm formation in *A. baumannii* (*Loehfelm, Luke & Campagnari, 2008*; *Gaddy, Tomaras & Actis, 2009*). Together, the regulation of biofilm formation is multifactorial, and many molecular mechanisms remain unidentified.

Although biofilm formation in bacteria has been studied extensively (*Kalkanci & TunÇCan, 2019*) and described in *A. baumannii* previously (*Kim et al., 2015*), the characteristics of biofilm production in clinical isolates from hospitals in Taiwan have seldom been well characterized. Therefore, we used the *A. baumannii* strains from our previous studies to examine the difference in biofilm formation among these strains with a modified XTT assay. In addition, we also closely observed biofilm formation using scanning electron microscopy (SEM) and confocal laser scanning microscopy (CLSM). These experiments were designed to understand the nature of biofilm production among the wild-type strain, mutant strains and clinical isolates of *A. baumannii*. To examine the impact of different compounds on biofilm formation and find potential candidates for anti-biofilm agents, different kinds of antibiotics, the antimicrobial peptide LL-37 and the iron chelating agent tannic acid, were chosen to test the relationship between their presence and biofilm production in *A. baumannii*. Finally, to understand how the presence of an *A. baumannii* biofilm influenced antimicrobial therapy, we determined the planktonic minimum inhibitory concentrations and minimum biofilm eradication concentrations. The result might help clinicians decide the optimal therapy while treating the infections caused by *A. baumannii* with a high biofilm-producing ability.

## MATERIALS & METHODS

### Bacterial strains

All of the bacterial strains used in this study were isolates from our previous studies (*Lin et al., 2014*; *Lin, Lin & Lan, 2015*; *Lin et al., 2015*; *Lin, Lin & Lan, 2017a*; *Lin et al., 2017b*) or American Type Culture Collection (ATCC), as shown in Table 1. The studied strains contained quality control strain ATCC 19606, ATCC 17978 and 21 clinical isolates. Five antibiotic-induced resistant strains from ATCC 17978 or tigecycline resistant clinical isolate ABhl1 were also included. Since biofilm formation is associated with antibiotic resistance and *baeR* as well as *emrA/emrB* have been shown related with tigecycline and colistin resistance respectively in our previous studies (*Lin et al., 2014*; *Lin, Lin & Lan, 2017a*), so we also enrolled the mutant strains for these genes.

### Biofilm quantification

Biofilm formation was performed in 24-well polystyrene plates as previously described with some modifications (*Marti et al., 2011*; *Orsinger-Jacobsen et al., 2013*; *Kim et al., 2015*). Biofilm formation was determined in Mueller Hinton Broth (MHB) using an initial $OD_{600}$ of 0.01, and the plates were incubated at 37 °C for 24 h without shaking (24-well, one mL). Two wells were left uninoculated and used as negative controls. The culture media was removed by inversion, and the wells were washed twice with phosphate buffered saline (PBS).

Both crystal violet and XTT assays show an excellent application for quantification of biofilms. The Crystal violet assay is cheap, easy and is usually used for the quantification of biofilms formed by microorganisms but XTT is more reliable and repeatable (*Hendiani, Abdi-Ali & Mohammadi, 2014*). In this study, XTT assay was used to measure the metabolic activity in order to estimate the burden of viable cells. The electron transport system in the cellular membrane of live bacteria reduces the XTT tetrazolium salt to XTT formazan, which results in a colorimetric change measured at 492 nm (*Orsinger-Jacobsen et al., 2013*). 250 μL XTT solution (0.0625 mg/mL XTT, 0.5 mM menadione in PBS) was added to each well of 24-well microtiter plate for 30 min at 37 °C. After incubation, 200 μL of the XTT supernatant was transferred to a fresh 96-well plate, and the calorimetric absorbance was measured ($OD_{490nm(XTT)biofilm}$) at 490 nm using iMarkTM Microplate Absorbance Reader (Bio-Rad laboratories, Inc. USA). The negative control ($OD_{490nm(XTT)control}$) was used to reduce the background absorbance optic density values in the XTT assay. Since different bacterial strains had different growth rates, the total cell numbers would be different at the end of incubation, thus influencing the total biofilm mass or metabolic activities. To correct the bias due to cell numbers, the $OD_{595nm}$ ($OD_{595nm(cells)}$) was determined to estimate the total cells in each corresponding well of 24-well microtiter plate after transferred to a new 96-well plate for optic density measurement. The ability to form a biofilm was expressed using a biofilm formation index: [$BFI = (OD_{490nm(XTT)biofilm} - OD_{490nm(XTT)control})/OD_{595nm(cells)}$]. The ODc was defined as three standard deviations above the mean OD of the negative control. Each isolate was classified as follows: non-biofilm producer: $OD \leq ODc$; weak biofilm producer: $ODc < OD \leq 2 \times ODc$; moderate biofilm producer: $2 \times ODc < OD \leq 4 \times ODc$; or strong biofilm producer: $OD > 4 \times ODc$ (*Hu et al., 2016*).

**Table 1  Bacterial strains used in this study.**

| Strain | Relevant feature(s) | Source or reference |
|---|---|---|
| 19606 | *A. baumannii* Wild-type strain | ATCC |
| 17978 | *A. baumannii* Wild-type strain | ATCC |
| Δ*baeR* | Derived from ATCC 17978. *baeR* mutant obtained by *kan*$^r$ gene replacement | *Lin et al. (2014)* |
| Δ*emrA* | Derived from ATCC 17978. A1S_1800 mutant obtained by *kan*$^r$ gene replacement | *Lin, Lin & Lan (2017a)* |
| Δ*emrB* | Derived from ATCC 17978. A1S_1772 mutant obtained not by *kan*$^r$ gene replacement | *Lin, Lin & Lan (2017a)* |
| Δ*ompA* | Derived from ATCC 17978. *ompA* mutant obtained by *kan*$^r$ gene replacement | *Lin et al. (2015)* |
| ABamk | Induced amikacin resistant ATCC 17978 | *Lin et al. (2017b)* |
| ABipm | Induced imipenem resistant ATCC 17978 | *Lin et al. (2017b)* |
| ABcol | Induced colistin resistant ATCC 17978 | *Lin et al. (2017b)* |
| ABtc | Induced tigecycline resistant ATCC 17978 | *Lin et al. (2014)* |
| ABtcm | Derived from ABtc. *baeR* mutant obtained by *kan*$^r$ gene replacement | *Lin et al. (2014)* |
| ABhl1 | Tigecycline resistant clinical isolate from Hualien Tzu Chi Hospital | *Lin et al. (2014)* |
| ABhl1tc | ABhl1 with induced high tigecycline resistance | *Lin, Lin & Lan, (2015)* |
| CT11-14 | MDRAB isolate from the Chut-Tung branch of National Taiwan University Hospital | *Lin et al. (2017b)* |
| CM1-4 | MDRAB isolate from Catholic Mercy Hospital | *Lin et al. (2017b)* |
| HC1-5 | MDRAB isolate from the Hsin-Chu branch of National Taiwan University Hospital | *Lin et al. (2017b)* |
| VGH1-7 | MDRAB isolate from the Hsin-Chu branch of Taipei Veterans General Hospital | *Lin et al. (2017b)* |

## Biofilm imaging

Biofilms were examined by SEM and CLSM (*Djeribi et al., 2012*). For SEM examination, biofilms were developed at 37 °C for 24 h in polystyrene coverslips placed into a 24-well plate with MHB containing bacterial cells (OD$_{600nm}$ 0.01) added. After fixed in 3.7% formaldehyde for 40 min at room temperature and then rinsed twice with PBS, further fixation was performed with 1% osmium tetroxide for 15 min at room temperature and then rinsed twice with PBS. The samples were then dehydrated by passing them through different concentrations of ethanol: 35%, 50%, 70%, 80%, and 95%, each for 5 min, followed by 100% ethanol twice for 10 min each time. The samples were then dried at 60 °C for 24 h. After being coated with gold-palladium (via sputter coating), the samples were examined under a bioscanning electron microscope (Hitachi, S-4700, Type II).

For CLSM examination, biofilms formation in polystyrene coverslips and fixation in 3.7% formaldehyde were the same as mentioned above. Then the samples were stained with a prepared 100 nM solution of Syto-9 stain (Invitrogen, Carlsbad, CA) for 30 min (*Luo et al., 2015b*). Syto-9 stained biofilms were excited with a 488 nm solid-state laser, and fluorescence was captured between 500–550 nm. The images of biofilms were rendered and assembled using appropriate computational software (ZEISS LSM780).

### Influence of antibiotics, LL-37 and tannic acid on biofilm formation

The activity of the various antibiotics, including amikacin, imipenem, colistin and tigecycline, LL-37 and tannic acid, on bacterial growth was tested in 24-well microtiter plates containing the above compounds at different concentrations. The concentrations chosen for different compounds were based on individual MIC with two-fold serial dilution. However, if biofilm formation was not observed at the higher concentrations of compounds, then the lower concentration was adopted. Biofilm production was checked by the same method as described above. Finally, *A. baumannii* ATCC 17978 without any compound added was used as a control for biofilm formation to calculate the relative biofilm formation capacity (relative biofilm formation index [RBFI] = $BFI/BFI_{A.baumannii\ ATCC\ 17978}$).

### Planktonic susceptibility test

A planktonic susceptibility test was performed as previously described to determine the minimum inhibitory concentration (MIC) of *A. baumannii* (*Lin et al., 2014*). In brief, bacteria were inoculated into one mL cation-adjusted Mueller-Hinton Broth (CAMHB) (Sigma-Aldrich, St. Louis, MO) containing different concentrations of antibiotics, including amikacin, imipenem, colistin and tigecycline, to reach $\approx 5 \times 10^5$ CFU/mL, and the cultures were incubated at 37 °C for 24 h. The lowest antibiotic concentration that completely inhibited bacterial growth was defined as the minimum inhibitory concentration, and growth was determined by unaided eyes and by measuring optical densities using a spectrophotometer.

### Biofilm susceptibility test

The biofilm susceptibility test was performed as previously described with some modifications (*Kim et al., 2015*). In brief, preparation of the antibiotic stock solution and preparation of the inoculum were the same as the method used in checking the MIC. Biofilm susceptibility tests were performed in flat-bottom, polystyrene cell-culture microtiter plates containing $5 \times 10^5$ CFU/mL in MHB with final well volumes of 100 µL. The plates were incubated at 37 °C without shaking. After 24 h, all the suspension fluid was removed from the plates and then washed with PBS. Equal amounts of appropriate antibiotic dilutions were added. Twofold dilutions of the tested antibiotics in CAMHB were diluted from 256 to 0.125 µg/mL. The tested antibiotic concentrations would be increased if necessary. Then, the plates were incubated at 37 °C for 24 h without shaking. After 24 h, inhibition of bacteria growth was determined by no turbidity observed by unaided eyes. Regrowth of bacteria was further determined by spot assay. The minimum biofilm eradication concentration (MBEC) represents the lowest drug concentration at which bacteria failed to regrow (*Olson et al., 2002*).

### Statistical analysis

The comparison of biofilm formation, including BFI and RBFI, was analyzed using Student's *t*-test. All of the data were from three independent experiments and analyzed using Statistical Package for the Social Sciences version 16 (SPSS Inc., Chicago, IL, USA). $p < 0.05$ was considered statistically significant.

## RESULTS

### Biofilm quantification and its relationship with the antimicrobial susceptibility

Biofilm quantification of the 33 different strains of *A. baumannii* (Table 1), including 21 clinical isolates, by the biofilm formation index is presented in Fig. 1. There are obvious differences in biofilm formation among the clinical isolates of *A. baumannii*. Of the 22 clinical isolates including *A. baumannii* ATCC 17978, twenty isolates were strong biofilm producers ($4 \times ODc < OD_{490nm}$) except VGH1 and CT11 ($2 \times ODc < OD_{490nm} \leq 4 \times ODc$) were moderate biofilm producers (data shown in the raw data set).

The antimicrobial susceptibility of the clinical isolates in this study has been published previously (*Lin et al., 2017b*) as shown in Table S1. Susceptibility breakpoints to antimicrobial agents were determined by the MIC interpretive standards for *Acinetobacter* spp. of Clinical and Laboratory Standards Institute (*CLSI, 2012*). The MIC interpretive criteria for each individual antimicrobial agents except cefazolin, cefmetazole and tigecycline (MIC µg/mL) (S = susceptible, I = intermediate and R = resistant) are listed respectively as the following: Ampicillin/sulbactam (S ≤ 8/4, I 16/8 and R ≥ 32/16); piperacillin/tazobactam (S ≤ 16/4, I 32/4 − 64/4 and R ≥ 128/4) ; cefotaxime (S ≤ 8, I 16-32 and R ≥ 64); ceftazidime (S ≤ 8, I 16 and R ≥ 32); cefepime (S ≤ 8, I 16 and R ≥ 32); imipenem (S ≤ 4, I 8 and R ≥ 16); meropenem (S ≤ 4, I 8 and R ≥ 16); amikacin (S ≤ 16, I 32 and R ≥ 64); gentamicin (S ≤ 4, I 8 and R ≥ 16); ciprofloxacin (S ≤ 1, I 2 and R ≥ 4); levofloxacin (S ≤ 2, I 4 and R ≥ 8); trimethoprim/sulfamethoxazole (S ≤ 2/38 and R ≥ 4/76). *A. baumannii* is naturally resistant to cefazolin, whereas the isolates of *A. baumannii* with cefmetazole MICs of ≥ 32µg/mL are considered resistant (*Jones et al., 1986*). The provisional MIC breakpoints for tigecycline are ≤2, 4 and ≥8 µg/mL to designate susceptible, intermediate and resistant strains, respectively (*Pachon-Ibanez et al., 2004*). All of the 21 *A. baumannii* clinical isolates are not susceptible to many of the tested antimicrobial agents. Although ATCC 17978 is the least resistant strain among the isolates studied, its BFI ranked second. Two isolates, VGH2 and VGH7, are pan-drug resistant. However, their biofilm formation ability BFI differed with VGH2 ranking first and VGH7 ranking sixteenth. It seems the capacity of biofilm production does not relate to antimicrobial susceptibility as shown in Table S1. To validate the inference, the correlation between biofilm formation ability and resistance to the 15 antimicrobial agents in *A. baumannii* was analyzed using the Wilcoxon rank-sum test. Figure 2 depicts the box plots with the comparison of the biofilm formation indices of the susceptible and resistant strains to the individual antibiotics. It reveals no relationship with statistical significance ($p > 0.05$) between the ability to form biofilm and the antimicrobial susceptibility of the *A. baumannii* clinical isolates.

### Biofilm formation by *A. baumannii* ATCC 17978 after *baeR*, *emrA*, *emrB* and *ompA* gene knockout

Many proteins, including two-component regulatory system BfmRS (*Luo et al., 2015a*), CsuA/BABCDE usher-chaperone assembly system (*Tomaras et al., 2008*), biofilm-associated proteins (*Loehfelm, Luke & Campagnari, 2008*), OmpA protein (*Gaddy,*

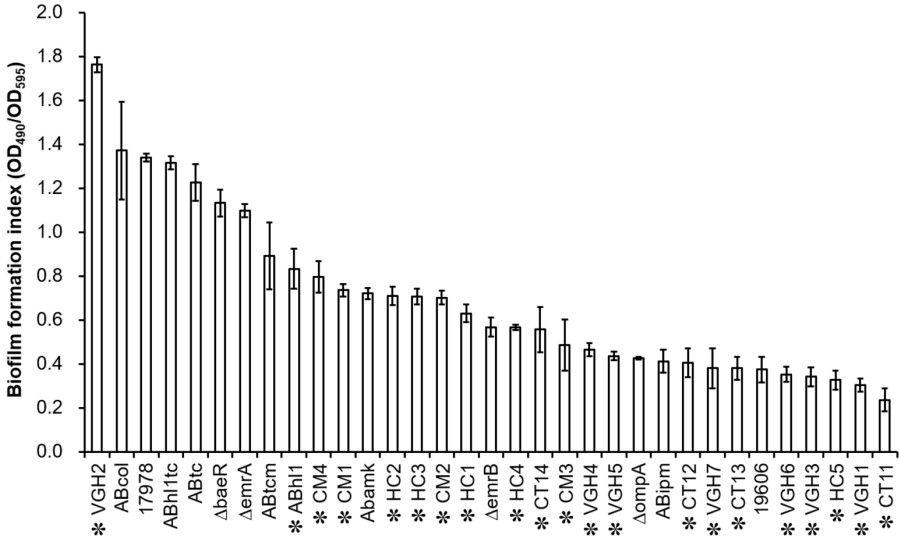

**Figure 1** **Biofilm quantification of the 33 different strains, including 21 clinical isolates of *A. baumannii* (highlighted by asterisk markers).** There are obvious differences in biofilm formation among the clinical isolates of *A. baumannii*.

*Tomaras & Actis, 2009*), AdeFGH (*He et al., 2015*) and *bla*PER (*Lee et al., 2008*), have been demonstrated to be related with biofilm formation in *A. baumannii*. To understand the scope of involvement further, biofilm formation caused by *A. baumannii* ATCC 17978 after gene knockout of another two-component regulatory system gene *baeR*, efflux pump genes *emrA/emrB* and *ompA* is compared as shown in Fig. 3A. The Δ*baeR*, Δ*emrA*, Δ*emrB* and Δ*ompA* mutant strains all had less biofilm formation than the wild type strain. This result is further confirmed by the SEM and CLSM images shown in Figs. 3B and 3C, respectively. Of these three categories of genes, *ompA* knockout led to the most marked decrease biofilm formation in *A. baumannii*. The SEM figure of Δ*ompA* strain shows that only few cells are clustered and its images of CLSM exhibit the thinnest biofilm. Although the influence on biofilm formation reduction after *baeR*, *emrA*, or *emrB* gene knockout was not so obvious as the Δ*ompA* strain, the degree of decrease was still statistically significant while being compared with the ATCC 17978 strain ($p < 0.05$ for Δ*baeR*, $p < 0.01$ for Δ*emrA*, and $p < 0.001$ for Δ*emrB*).

## Influence of different compounds on biofilm formation

Table 2 presents minimum inhibitory concentrations (MICs) of *A. baumannii* ATCC 17978 and VGH2 to amikacin, imipenem, colistin, tigecycline, LL-37 and tannic acid. The influence of different compounds, including amikacin, imipenem, colistin, tigecycline, LL-37 and tannic acid, on biofilm formation in *A. baumannii* ATCC 17978 and VGH2 is shown in Fig. 4 and Fig. 5. Amikacin, imipenem, colistin and LL-37 were tested at 1/4, 1/2 and 1 x MIC whereas tigecycline was tested at 1/16, 1/8, 1/4 x MIC. Since tannic acid had very high MIC (>300 μg/mL), the concentrations of 200, 100 and 50 μg/mL were chosen arbitrarily for testing. The addition of amikacin (>0.5 μg/mL), colistin (>0.5

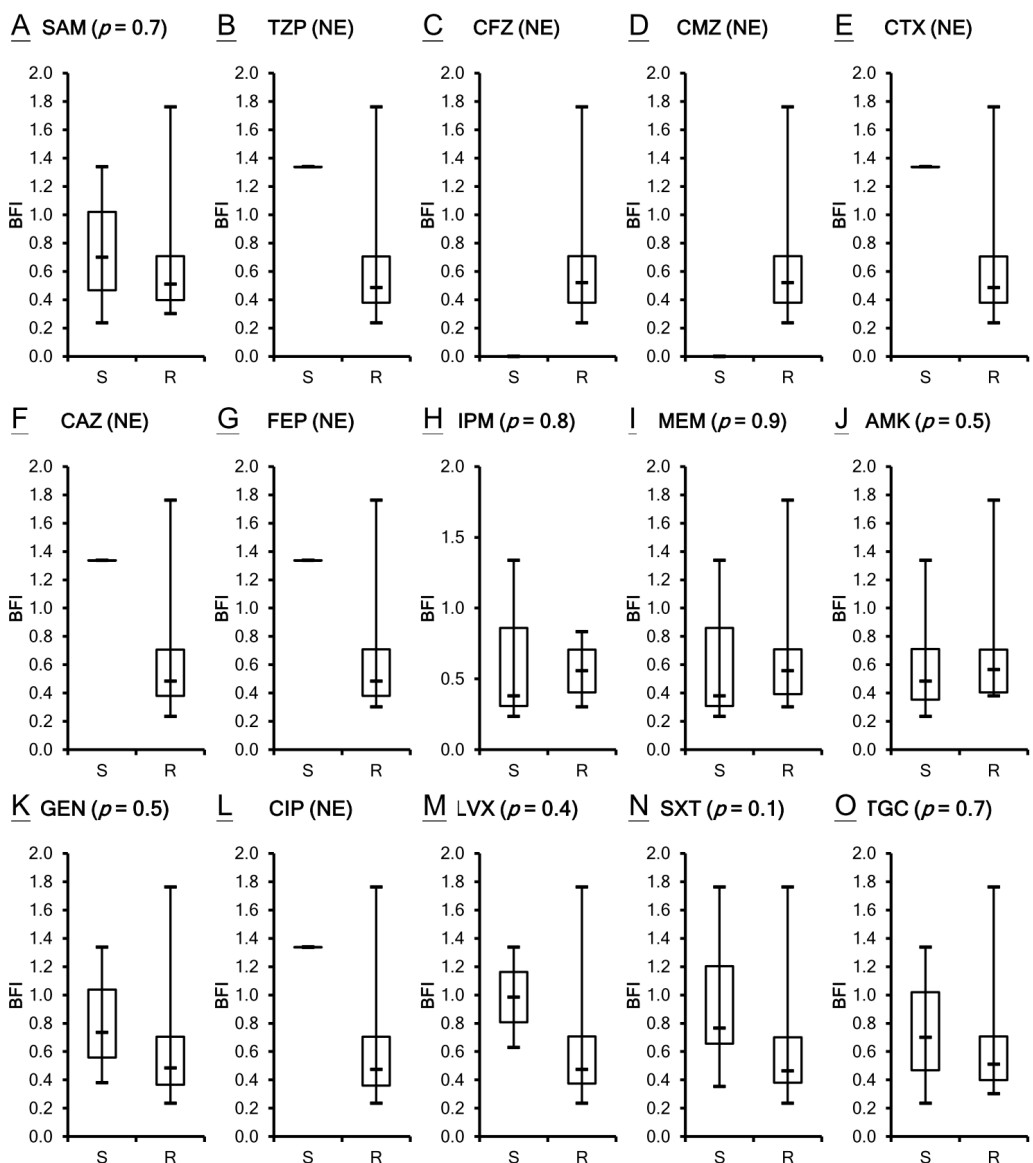

**Figure 2  Comparison of the biofilm formation indices of the susceptible and resistant strains to the individual antibiotics among *A. baumannii* ATCC 17978 and 21 clinical isolates.** (A–O) The antimicrobial susceptibility of *A. baumannii* isolates was mainly determined according to the MIC interpretive criteria published by the CLSI in 2012 (M100-S22). The MICs of individual antibiotics against the clinical isolates were shown in Table S1. No relationship with statistical significance was demonstrated between the ability to form biofilm and the antimicrobial susceptibility of the *A. baumannii* clinical isolates. The Student's *t*-test ($p < 0.05$) was used to determine the statistical significance of the experimental data. Abbreviations: SAM, ampicillin/sulbactam; TZP, piperacillin/tazobactam; CFZ, cefazolin; CMZ, cefmetazole; CTX, cefotaxime; CAZ, ceftazidime; FEP, cefepime; IPM, imipenem; MEM, meropenem; AMK, amikacin; GEN, gentamicin; CIP, ciprofloxacin; LVX, levofloxacin; SXT, trimethoprim/sulfamethoxazole; TGC, tigecycline; NE, not evaluated; S, susceptible; R, resistant; BFI, biofilm formation index.

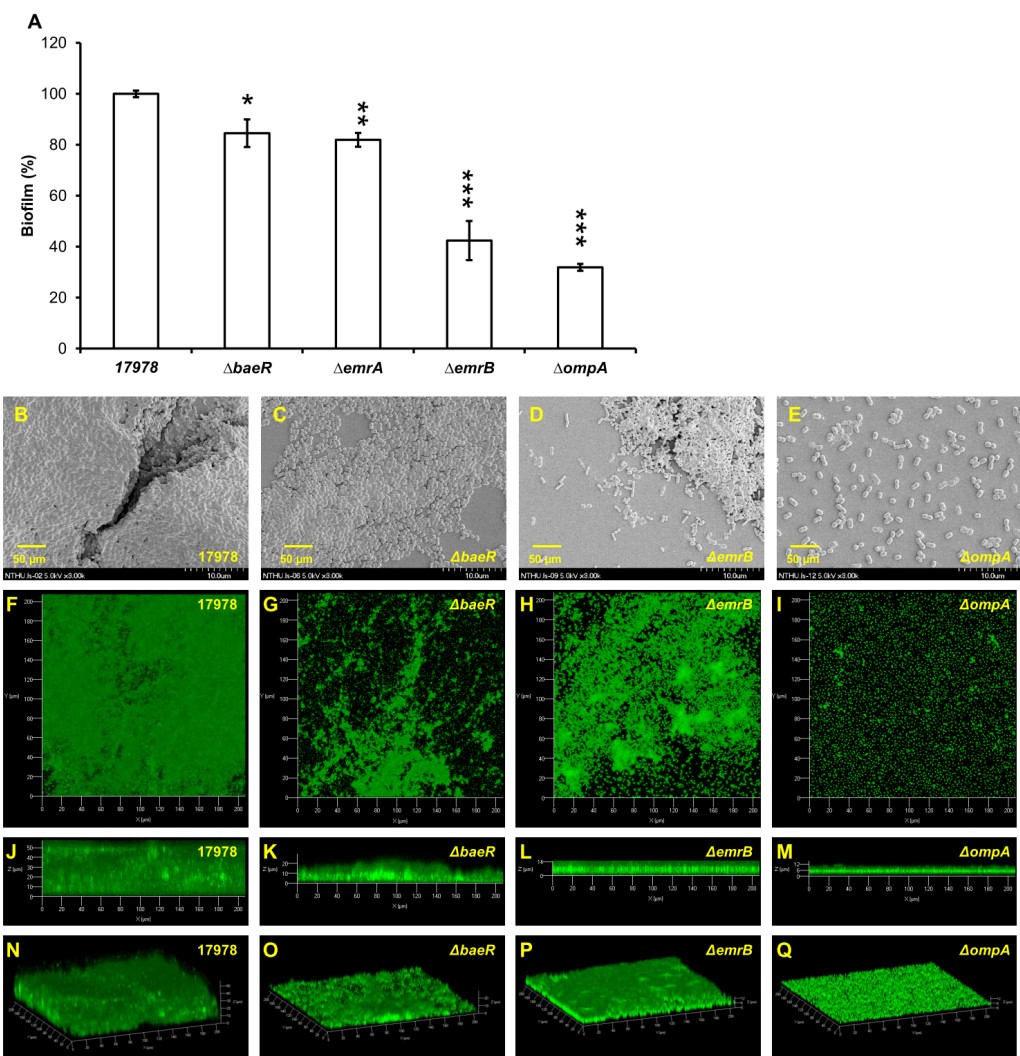

**Figure 3** **Biofilm formation by *A. baumannii* ATCC 17978 after gene knockout; (A) comparison between the wild-type stain and its mutants (B–E) SEM and (F–Q) CLSM images.** Δ*baeR*, Δ*emrA*, Δ*emrB* and Δ*ompA* all had less biofilm formation than the wild type. The results are shown as the means ± SD from three independent experiments. The Student's *t*-test was used to determine the statistical significance of the experimental data. *, $p < 0.05$ and **, $p < 0.01$ and ***, $p < 0.001$ between ATCC 17978 and the gene knockout mutant strains.

µg/mL), LL-37 (>4 µg/mL), or tannic acid (>50 µg/mL) decreased the biofilm formation ability of *A. baumannii* ATCC 17978, whereas the addition of amikacin (>256 µg/mL), imipenem (>4 µg/mL), colistin (>0.125 µg/mL), LL-37 (>2 µg/mL), or tannic acid (>50 µg/mL) decreased the biofilm formation ability of VGH2. In contrast, the addition of tigecycline increased the biofilm formation ability of *A. baumannii* ATCC 17978 (>62.5 ng/mL) and VGH2 (>0.5 µg/mL). Figure 6 presents the SEM and CLSM images of biofilm formation by *A. baumannii* ATCC 17978 in the presence of 1 µg/mL colistin or 125 ng/mL tigecycline. With 1 µg/mL colistin added to the ATCC 17978 strain, the SEM image shows

**Table 2  MIC and MBEC of *A. baumannii* ATCC 17978 and VGH2.**

|  | ATCC 17978 | | VGH2 | |
|---|---|---|---|---|
|  | **MIC** | **MBEC** | **MIC** | **MBEC** |
| Amikacin (µg/mL) | 1 | 32 | 1024 | >8192 |
| Imipenem (µg/mL) | 0.25 | 1 | 16 | 512 |
| Colistin (µg/mL) | 1 | >8192 | 0.5 | >8192 |
| Tigecycline (µg/mL) | 0.5 | 2 | 4 | 32 |
| LL37 (µg/mL) | 16 | ND | 8 | ND |
| Tannic acid (µg/mL) | >300 | ND | >300 | ND |

**Notes.**
ND, Not detected.

the cells become scattered and the CLSM images exhibit decreased thickness of biofilm. In contrast, 125 ng/mL tigecycline exposure has the opposite impact on biofilm formation of *A. baumannii* ATCC 17978. Both of the SEM and CLSM images validated the findings of biofilm quantification by RBFI analysis (Figs. 4 and 5).

## MICs and MBECs of *A. baumannii* ATCC 17978 and VGH2

Besides MICs, MBECs of *A. baumannii* ATCC 17978 and VGH2 to amikacin, imipenem, colistin and tigecycline is also shown in Table 2. In this study, the MIC values for antimicrobials were determined using a broth microdilution assay. The ATCC 17978 strain had low MICs to all of the four tested antimicrobial agents, whereas the VGH2 strain had higher MICS for amikacin (1,024 µg/mL) and imipenem (16 µg/mL). When biofilms cells of the same strains were tested, MBECs rose astonishingly with at least fourfold higher relative to the planktonic MICs. Of all the four antibiotics tested, colistin had the most obvious MBEC increase from 0.5 to >8,192 µg/mL for VGH2. This result suggested that biofilm cells of bacteria would become more resistant to tested antibiotics according to MBEC.

## DISCUSSION

High variability in biofilm formation exists among the clinical isolates of *A. baumannii* (*Wroblewska et al., 2008*). In a previous study that examined 86 clinical isolates of *A. baumannii*, clinical isolates exhibited an enhanced biofilm formation ability relative to a standard *A. baumannii* strain (ATCC 19606) (*Kim et al., 2015*). Interestingly, only *A. baumannii* VGH2 has better biofilm formation ability than the standard *A. baumannii* strain (ATCC 17978) in this study. However, Kim et al. used crystal violet to measure the biomass of biofilm whereas we measured the metabolic activity in order to estimate the burden of viable biofilm cells by tetrazolium salt XTT reduction assay instead because the XTT assay is the test most commonly used to estimate viable biofilm growth and to examine the impact of biofilm therapies (*Nett et al., 2011*). One previous study about candidal biofilm showed the better reproducibility of the XTT assay compared with that of the crystal violet method led the authors to conclude that the former is more reliable (*Jin et al., 2003*). However, using a metabolic activity assay for measuring antibiotic activity

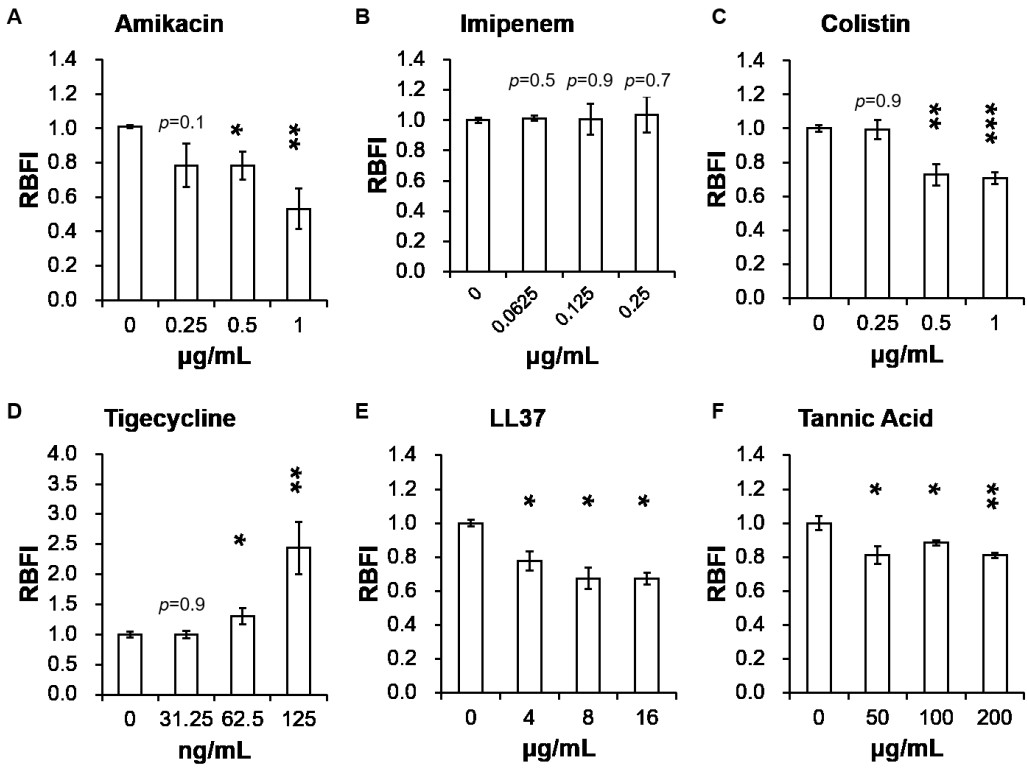

**Figure 4** **Influence of different compounds on biofilm formation in *A. baumannii* ATCC 17978; (A) amikacin, (B) imipenem, (C) colistin, (D) tigecycline, (E) LL-37, and (F) tannic acid.** The addition of amikacin ($> 0.5\,\mu$g/mL), colistin ($> 0.5\,\mu$g/mL), LL-37 ($> 4\,\mu$g/mL), or tannic acid ($> 50\,\mu$g/mL) decreased the biofilm formation ability of *A. baumannii* ATCC 17978, whereas the addition of tigecycline ($> 62.5$ ng/mL) increased the biofilm formation ability of *A. baumannii* ATCC 17978. The Student's *t*-test was used to determine the statistical significance of the experimental data. *, $p < 0.05$ and **, $p < 0.01$ and ***, $p < 0.001$ between *A. baumannii* ATCC 17978 with and without chemical compounds.

against biofilm suffers from the limitation that usually the absorbance or fluorescence signal decreases in proportion to the number of bacteria only on a limited range of colony forming units (CFUs). Statistical analyses showed that XTT activity was linearly associated with the log of the cell concentration over the cell concentration range tested (from $2 \times 10^5$ to $10^8$ cells/ml), implying XTT activity can be used as an indicator of cell numbers in a cell suspension (*Jin et al., 2003*; *Orsinger-Jacobsen et al., 2013*). Although we didn't determine the zone of linearity in this study, all of the absorbance OD$_{595}$ values, even in the presence of subinhibitory antibiotic concentrations, were more than 0.1 (data shown in the raw data set), which indicated that cells were present more than $10^5$. Therefore, the XTT assay in this study was presumed to be applicable in measuring the biofilm formation of *A. baumannii*. On the other hand, major differences in metabolic activity can be observed when comparing several bacterial isolates from the same species (*Zimmermann et al., 2015*), so that differences in XTT signal could rely simply on difference in metabolic activity. In this study, the influence of chemical compounds on biofilm formation of ATCC 17978 or

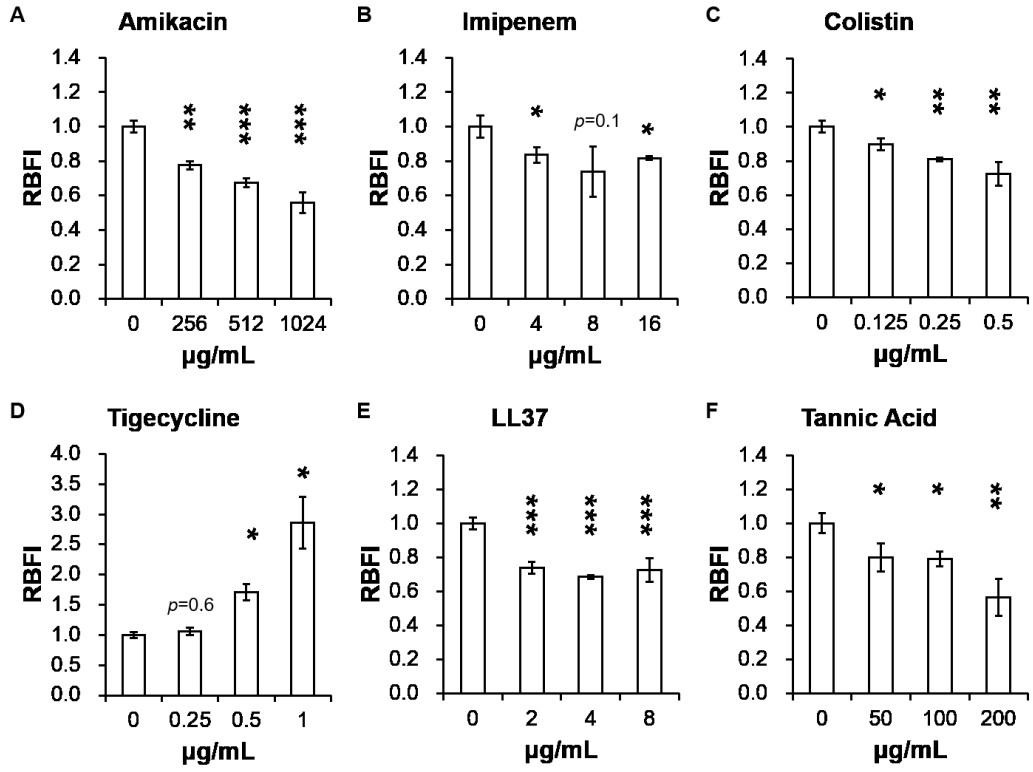

**Figure 5** **Influence of different compounds on biofilm formation in *A. baumannii* VGH2; (A) amikacin, (B) imipenem, (C) colistin, (D) tigecycline, (E) LL-37, and (F) tannic acid.** The addition of amikacin ($> 256 \, \mu\text{g/mL}$), imipenem ($> 4 \, \mu\text{g/mL}$), colistin ($> 0.125 \, \mu\text{g/mL}$), LL-37 ($> 2 \, \mu\text{g/mL}$), or tannic acid ($> 50 \, \mu\text{g/mL}$) decreased the biofilm formation ability of VGH2, whereas the addition of tigecycline ($> 0.5 \, \mu\text{g/mL}$) increased the biofilm formation ability of VGH2. The Student's $t$-test was used to determine the statistical significance of the experimental data. *, $p < 0.05$ and **, $p < 0.01$ between *A. baumannii* VGH2 with and without chemical compounds.

VGH2 was compared to itself. Therefore, this phenomenon would not change the main findings of this study.

In this study, all of the 21 *A. baumannii* clinical isolates could form biofilms. However, the clone relatedness and the ability of biofilm formation were not further explored. One previous study reported higher ability of biofilm formation was found for the strains assigning to multilocus sequence typing (MLST) sequence type 2 (ST2), ST25 and ST78 (*Giannouli et al., 2013*). But in another study the sporadic strains showed significantly higher biofilm-forming capacity than the epidemic isolates assigning to ST2 (*Hu et al., 2016*). It seems the biofilm-forming capacity in various sequence types of *A. baumannii* needs further experiments to clarify.

A few studies have investigated the role of RND efflux pumps, including AdeABC, AdeIJK and AdeFGH, in *A. baumannii* biofilm formation (*He et al., 2015*; *Yoon et al., 2015*). The role of efflux pumps in *A. baumannii* biofilm formation has been suggested in whole transcriptome analysis of biofilm and planktonic cells (*Rumbo-Feal et al., 2013*), although EmrAB was not implicated in gene expression change of biofilm cells. However,
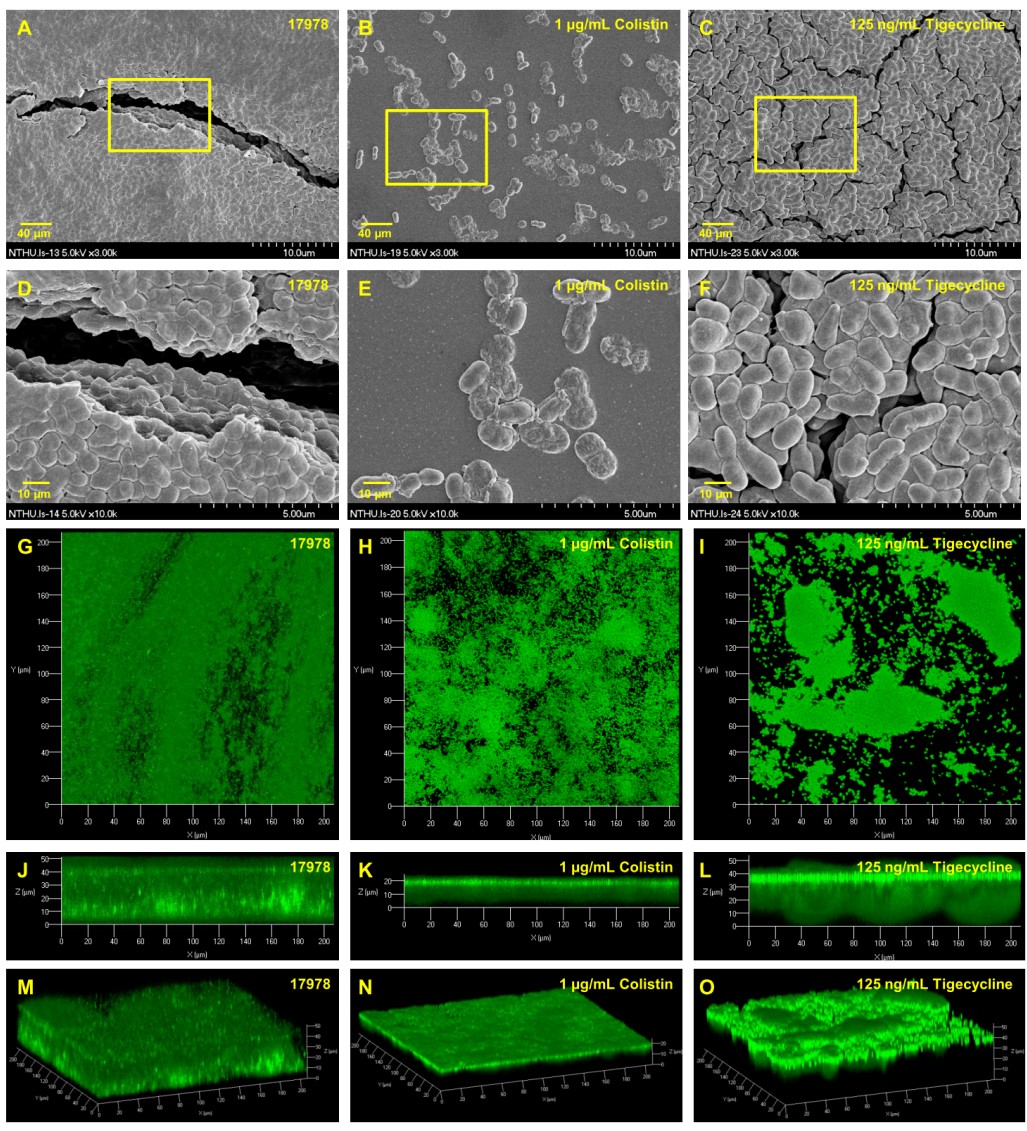

**Figure 6  SEM and CLSM images of biofilm formation by *A. baumannii* ATCC 17978 after colistin and tigecycline challenge, (A-F) SEM and (G-O) CLSM.** The SEM image shows the cells become scattered and the CLSM images exhibit decreased thickness of biofilm. In contrast, 125 ng/mL tigecycline exposure has the opposite impact on biofilm formation of *A. baumannii* ATCC 17978.

*emrA* and *emrB* genes have been demonstrated related with biofilm formation in other bacteria. *Escherichia coli* mutant strains lacking the efflux *emrA* and *emrB* were reported to display enhanced biofilm growth compared with control (*Bay et al., 2017*),where the deletion of the efflux genes *emrAB* in *Salmonella enterica serovars* resulted in decreased biofilm formation compared with wild type strain (*Baugh et al., 2012*). In this study, the efflux pump EmrAB was found to be associated with biofilm formation in *A. baumannii* for the first time. Although we have demonstrated that EmrAB contributes to adaptation

to osmotic stress and resistance to colistin in *A. baumannii* (*Lin, Lin & Lan, 2017a*), its role in biofilm formation still needs to be clarified.

Besides *emrA*, *emrB* and *ompA* genes implicated in biofilm formation of *A. baumannii* in this study, deletion of two-component system gene *baeR* led to decreased biofilm formation. Deleting *baeR* of *A. baumannii* results in vulnerable to certain chemicals, especially tigecycline and tannic acid (*Lin et al., 2014*; *Lin, Lin & Lan, 2015*). In addition to involvement of disposing chemicals, two-component signal transduction system is also regarded as a major strategy for connecting input stimuli to biofilm formation (*Liu et al., 2018*). The current proposed model associates low intracellular levels of c-di-GMP with a planktonic lifestyle, whereas high c-di-GMP levels are associated with biofilm formation. However, it has been demonstrated that the BaeSR response in *E. coli* does not influence biofilm formation, nor is it involved in indole-mediated inhibition of biofilm formation (*Leblanc, Oates & Raivio, 2011*). To investigate the mechanism for biofilm formation decrease after *baeR* deletion and clarify the true role of BaeSR in biofilm formation of *A. baumannii*, a secondary messenger c-di-GMP concentration in *A. baumannii* needs to be measured in the future research since c-di-GMP is a well-known intracellular messenger molecule that affects biofilm formation.

Exposure to aminoglycoside and subinhibitory concentrations of imipenem is associated with biofilm-forming *A. baumannii* isolates (*Rodríguez-Baño et al., 2008*; *Nucleo et al., 2009*). In two pairs of clinical colistin-susceptible/colistin-resistant (Csts/Cstr) *A. baumannii* strains, the Cstr strains showed significantly decreased biofilm formation in static and dynamic assays ($p < 0.001$) and lower relative fitness ($p < 0.05$) compared with those of the Csts counterparts (*Dafopoulou et al., 2015*). We found that *A. baumannii* ATCC 17978 decreased biofilm formation upon exposure to subinhibitory concentrations of amikacin, imipenem and colistin. In this study, tannic acid and LL-37 at a concentration below its MIC can decrease biofilm formation, which might occur by causing structural damage to the *A. baumannii* biofilm (*Shi et al., 2014*). One previous study has demonstrated increased biofilm formation of *Staphylococcus epidermidis* after sub-MIC tigecycline treatment (both at 0.25 and 0.5 MIC) by producing increased expression of the *icaA* (production of transmembrane protein), *altE* (encoding autolysin related with adhesion) *and sigB* (biofilm stability) genes and by affecting biofilm architecture in the isolates (*Szczuka, Jablonska & Kaznowski, 2017*). To explore the mechanism by which sub-MIC tigecycline increased the biofilm formation of *A. baumannii*, more future relevant studies are needed.

*Wang et al. (2016)* have shown that the minimal bactericidal concentrations for biofilm-embedded cells of the tested isolates were more than 50-fold higher than those for their planktonic cells. The lowest biofilm inhibitory concentration (BIC) value of clinical multidrug resistant *A. baumannii* (MDRAB) isolates for colistin was 32-fold the $MIC_{90}$ (minimum inhibitory concentration for 90% of the isolates) value, and $BIC_{90}$ values were found to be 512-fold the $MIC_{90}$ value (*Milletli Sezgin, Coban & Gunaydin, 2013*). This result is compatible with those in this study. Because treatment for biofilm-forming MDRAB infection has become more difficult, several regimens have been proposed as anti-biofilm treatment for *A. baumannii* infection. Treatment with imipenem and

rifampicin individually or in combination has obvious anti-biofilm effects (*Wang et al., 2014*). Song et al. demonstrated that tigecycline and colistin-rifampicin are effective for the prevention of or reduction in biofilm formation caused by *A. baumannii* strains (*Song & Cheong, 2015*). However, the influence of tigecycline on biofilm formation of *A. baumannii* is contrary to the results of this study. Meropenem plus sulbactam exhibited synergism against biofilm-embedded carbapenem-resistant *A. baumannii* and caused significantly more damage to biofilm architecture than any of the agents used alone (*Wang et al., 2016*).

The images of SEM and CLSM were used in this study to support the findings by XTT assay about the impact of gene knockout and adding antibiotics on biofilm formation, but no live-dead stain was performed. Live-dead stain consists of SYTO9, which stains live cells green, and propidium iodide (PI), which stains dead cells red (*Richmond et al., 2016*). This allows live versus dead cells to be detected and is more informative. Although we made an effort to characterize biofilm production of *A. baumannii* in different conditions, there were still some limitations in this study. As previously described, the interplay of these related genes in biofilm formation of *A. baumannii* and key secondary messenger c-di-GMP determination were not explored further. Besides, because the blood or tissue concentrations produced by infusions of the studied antimicrobials were not determined, it is hard to infer that the in vitro results will apply to the in vivo condition.

## CONCLUSIONS

In this study, the efflux pump gene *emrA/emrB* and two-component system gene *baeR* were found to be associated with biofilm formation and sub-MIC tigecycline led to increased biofilm formation in *A. baumannii* for the first time. In conclusion, the biofilm formation ability of *A. baumannii* was diverse in different strains, involved many genes and is influenced by many chemical compounds. The minimum biofilm eradication concentrations of all the tested antibiotics were increased for both the wild-type and clinical isolate of multidrug resistant *A. baumannii* VGH2.

### Funding
This research was funded by the National Taiwan University Hospital Chu-Tung Branch, grant number 106001. The funders had no role in study design, data collection and analysis, decision to publish, or preparation of the manuscript.

### Grant Disclosures
The following grant information was disclosed by the authors:
National Taiwan University Hospital Chu-Tung Branch: 106001.

### Competing Interests
The authors declare there are no competing interests.

## Author Contributions

- Ming-Feng Lin and Chung-Yu Lan conceived and designed the experiments, analyzed the data, authored or reviewed drafts of the paper, and approved the final draft.
- Yun-You Lin performed the experiments, analyzed the data, prepared figures and/or tables, and approved the final draft.

## Data Availability

The raw data is available in the Supplemental File.

## Supplemental Information

Supplemental information for this article can be found online at http://dx.doi.org/10.7717/peerj.9020#supplemental-information.

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
