# Peer review of "Characterization of biofilm production in different strains of Acinetobacter baumannii and the effects of chemical compounds on biofilm formation"

_PeerJ, doi:10.7717/peerj.9020_

## Round 0.1 · original submission · Major Revisions

As you can see, your paper triggered extensive and constructive remarks from the reviewers. While both considered your work as interesting and worthwhile to be published, both also thought that a major revision was needed and both provided useful suggestions in this context. If you are ready to perform the necessary work, I may consider your work again. If you choose to submit a revised version, please include a detailed and point by point rebuttal in which you explain where and how you have taken each reviewer's comment into account. If you disagree with some of these comments or suggestions, explain why. Please, note that your revised version will undergo a new round of review by the same or by different reviewers. I cannot, therefore, make any commitment about a final acceptance of your revised version.

Reviewer 1 ·

Basic reporting

1. Description of strains is poor. The authors just refer to previous publications, but table 1 is not self-explanatory. What do mean the acronyms CM, CT, HC , etc ? Another factor that could affect biofilm formation could be clonal relatedness. This has not been explored here.
2. There is no justification for the selection of genes that were studied.
3. The MICs (line 220) should be given before describing the effects of treatments on biofilms formation, otherwise, the choice of concentrations is unclear. This choice should be justified in the corresponding paragraph (which is not the case now). It is unclear to me why amikacin, imipenem, and colistin were tested at ¼, ½ and 1 x MIC while tigecycline was tested at 1/16, 1/8, ¼ x MIC.
4. Figure 1 : the 2 panels are redundant. It is easy to highlight in the top panel which strains are clinical isolates.
5. Figure 2: please mention which breakpoints were used to define S and R.
6.Table 2 is not very informative without breakpoints values; it is also partially redundant with Figure 2.
7.The abstract could be improved taking into account these comments but also making in more self-explanatory: what are the genes studied coding for, what is strain VGH2 ?
8. Lines 117-122: this paragraph should come earlier, when describing CLSM. Live-dead staining could have been more appropriate that syto-9, because more informative.
9. Line 151-152: what do the authors mean by ‘the presence of biofilm formation’ in a paragraph where they study the destructive effect of treatments on preformed biofilms ?
10. Line 171: what do the authors mean by ‘100% non-susceptible’ ?
11. Line 278-279: I do not see to which results this sentence is referring: ‘the induced colistin-resistant A. baumannii has better biofilm formation ability than the parent’
12. There are two references Lin et al, 2017. Please make the distinction between them by adding a and b.
13. Please check raw data for figure 2. There are some local language characters and some missing values.

Experimental design

1. Biofilms were quantified using an XTT assay. This is not a standard technique to evaluate biofilm formation. XTT is indeed rather a measure of metabolic activity in living cells than a measure of biofilm formation, which is classically evaluated using crystal violet staining (evaluation of the whole biomass). Additional comments on this method are:
a) Major differences in metabolic activity can be observed when comparing several bacterial isolates from the same species, so that differences in XTT signal could rely simply on difference in metabolic activity. This metabolic activity should therefore be evaluated for each strain in planktonic culture in order to determine if it is comparable among strains.
b) Using a metabolic activity assay for measuring antibiotic activity against biofilm suffers of the limitation that usually the absorbance or fluorescence signal decreases in proportion of the number of bacteria only on a limited range of CFUs. This zone of linearity should be determined to know the limit of sensibility of the assay.
c) The way BFI is calculated is very unclear to me. I do not see how the measure of the absorbance in the supernatant is a measure of biofilm OD and why BFI is thus a ratio between the OD of the supernatant and OD of the cells.
d) Line 165: how was determined the cutoff value of 0.5?

2. Testing of antibiotic activity against biofilms were performed in 96-well plates while the comparison of biofilm forming capacity and the effect of antibiotics on biofilm formation were performed in 24- well plates. How does this difference in support influence biofilm formation ?

Validity of the findings

1. Lines 228-232: I am not sure this last sentence is correct. It seems to depend on the antibiotic. For colistin and tigecycline, similar difference between MIC and BMIC (+/- 1 doubling dilution) were observed against both stains.
2. First paragraph of the discussion: I am afraid the authors cannot compare biofilm formation in their work and in other studies in which the authors compared different strains using crystal violet absorbance (see my comment #1 here above).
3. Second paragraph of the discussion: It is not needed to describe at length the role of genes that were not studied here in biofilm formation. It would be more useful to discuss the role of the genes studied (see also my comment #4 here above).

Additional comments

In this work, the authors compare the ability of a series of clinical isolates and of mutants from a reference strain of A. baumannii to form biofilm as well as to respond to antibiotics in these biofilms.
I identified major methodological issues and also think that the number of exhibits can be reduced, some of the information presented being redundant.

Reviewer 2 ·

Basic reporting

I made my comments below.

Experimental design

I made my comments below.

Validity of the findings

I made my comments below.

Additional comments

Dear Editor,

The paper is very comphrensive, too much detailed and interesting but needs major revision. Furthermore, it is a bit long (I am not sure if length is a problem for the journal).


My requests from the authors are below:
*methods please add references to subtitles/paragraphs without references*please add this well written review to references http://mjima.org/abstract.php?id=173

*in 24-well microtiter plates containing the above compounds at different concentrations
what concentrations please mention furthermore what were the mics of these commpounds for study strains


*LL-37 and tannic acid: why not planktonic susceptibility test not performed

*The data were analyzed using Statistical Package for the Social Sciences version 16 (SPSS Inc.,
158 Chicago, IL, USA). P < 0.05 was considered statistically significant.what rests were used and when?

*antibiogram change as antibiotic susceptibility test every part of the manuscript
*If 0.5 was assigned as the cutoff value for the strong and weak biofilm-forming abilities, we found that

please move this part to methods and revise the part in results

*give antibacterial susceptibility data of all study strains

*It reveals no relationship with statistical significance (P>0.05) between
please give exact pp everywhere in the manuscript
*how is the biofilm formation effected by the mean blood concentrations or tissue concentrations produced by standard infusions of your study drugs
*In this study, tannic acid and LL-37 at a concentration below its MIC can decrease biofilm formation,
I could not see data of iyt in results.
*add why the editor should publish your paper-new things in the paper and limitations before conclusion

---

## Round 0.2 · Major Revisions

One of the original reviewer commented that the revision was very superficial and that you did not answer to some of her criticisms. This is regrettable because I made very clear that you had to take all comments into consideration and, if you disagree with some of them, tell us why.
Thus, your paper definitely needs further revision. Please, work on this very carefully as it will be your last chance to have your data published in PeerJ.

Reviewer 1 ·

Basic reporting

The way the authors dealt with reviewers' comment is very scholar, no specific effort has been made to integrate the remarks in the body of the text or in the light of existing literature. A good example is the last sentence of the discussion regarding the limitations of the study.

Some of my questions have not at all been addressed. It seems that the authors did not understand what was meant.
Q1, related to clonal relatedness: it was obvious that I was asking the question for clinical isolates, and not for reference strains.
Q6: my question was related to the committee used as a reference for breakpoint values: EUCAST, CLSI, other ?
Q10: 100% non susceptible makes no sense as stated before. I guess that 100% refers to the fact that all isolates (100% of them) are non-susceptible. Therefore there is no need to repeat it. This sentence should probably read: All of the 22 A. baumannii clinical isolates except ATCC17978 are non-susceptible to many of the tested antimicrobial agents.

Experimental design

Again, some of my comments were not considered.

Q1 a) The authors add a sentence in the text to state that metabolic activity was determined in planktonic cultures (line 361), but they do not show the results.
b) they still do not provide the zone of linearity of they assay.
c) I still do not understand why they measure absorbance in cells and biofilms to estimate biofilm formation.
d) an arbitrary cut-off is of very poor predictive value and does not allow comparison with other studies.

Validity of the findings

In the absence of appropriate controls for the experiments related to biofilms and, more specifically, the XTT assay (as suggested above) , I am still doubtful about the value of these data.

---

## Round 0.3 · Minor Revisions

As you will see, the reviewer has now commented on your revision. Some additional corrections still need to be made. Please, pay a particular attention to these comments and spend the time and effort required to arrive at a final version. Once we get it, I'll re-examine your whole paper and hope to be able to convey a favourable opinion. However, I cannot make any commitment at this stage as my opinion on the suitability for publication of your submission will be dependent on your rebuttal and the revised version. I kook forward to hear from you in due course.

Reviewer 1 ·

Basic reporting

Many of the replies remain very scholar , the way the text is modified is not at all elegant.
If concentrating on the science
Q1. It is not acceptable to 'speculate' on the fact isolates could be different simply because they show from 5 hospitals (but there are 21 isolates, meaning a mean of 4 isolates per hopsital) and show different susceptibility patterns.
Q6. I was asking which susceptibility breakpoints were used, the authors reply be mentioning the method they used for measuring MICs

Experimental design

Q1 -c: please make clear in the formula the wavelength at which OD are measured, it will help understanding that the ODcells is measured at another wavelength that the XTT .
Q1-d: this is still unclear, as you removed the sentence on the cutoff at the first occurrence and not at the second one. The sentence you left at the first occurrence is not understandable. I think you did not understand my remark. I stated that setting an arbitrary cut off does not allow comparing your data with others. The question was whether the collection could not be divided in weak and strong biofilm producers using a statistical test allowing to split the biofilm values on a non-arbitrary basis. Moreover, I am not really sure a cutoff is really needed in the study.

Validity of the findings

no further comments

---

## Round 0.4 · Minor Revisions

Thank you for unwavering and further modifying your paper following the remarks of reviewer #1. I read you replies and this 3d version and find it quite acceptable. However, I think you still need to make one more correction without which your paper may be difficult to understand. It is about the breakpoints and the MICs you used. You indicated which interpretive criteria you used by referring to a rather old version of the CLSI pertinent document and then stated in your reply that you thought unimportant to indicate the MIC values. I disagree and think this is mistake and that you *should* indicate which MIC values were used (zones diameters need not be reported, as these are secondary values derived from the MIC).

For your information, I explain here why actual MIC values should be indicated in your paper. First these values are essential for a correct interpretation of resistance and susceptibility to an antibiotic (not the whole story but an important part). Since you used values that were changed in 2014, this will create confusion as most lay readers will use not the old 2012 document bu the new one (2020 !) and will miss the point. Also (i) since 2006, it is no longer the CLSI but the FDA that sets breakpoints for the US. In Europe, breakpoints are set by EUCAST. FDA and EUCAST breakpoints are often different from those of CLSI, increasing the confusion for non-speciaiists. So, the best way is to tell which MIC you have taken, whihc is a factual information.

Please, correct your submission and prepare a new version. I look forward receiving it.

---

## Round 0.5 · accepted · Accept

Thank you for providing the reader with the MIC values that you used to categorize your isolates as susceptible, intermediate, or resistant. I guess this will enable the reader to better understand your message and will spare her/him the effort at looking for a document published in 2012 (CLSI - Performance Standards for Antimicrobial
Susceptibility Testing) that is not readily available unless subscribed for.